# Estimating the heritability of SARS-CoV-2 susceptibility and COVID-19 severity

Kathleen LaRow Brown [1], Vijendra Ramlall[1,2], Michael Zietz [1], Undina Gisladottir [1] & Nicholas P. Tatonetti [1,3,4] ✉

SARS-CoV-2 has infected over 340 million people, prompting therapeutic research. While genetic studies can highlight potential drug targets, understanding the heritability of SARS-CoV-2 susceptibility and COVID-19 severity can contextualize their results. To date, loci from meta-analyses explain 1.2% and 5.8% of variation in susceptibility and severity respectively. Here we estimate the importance of shared environment and additive genetic variation to SARS-CoV-2 susceptibility and COVID-19 severity using pedigree data, PCR results, and hospitalization information. The relative importance of genetics and shared environment for susceptibility shifted during the study, with heritability ranging from 33% (95% CI: 20%-46%) to 70% (95% CI: 63%-74%). Heritability was greater for days hospitalized with COVID-19 (41%, 95% CI: 33%-57%) compared to shared environment (33%, 95% CI: 24%-38%). While our estimates suggest these genetic architectures are not fully understood, the shift in susceptibility estimates highlights the challenge of estimation during a pandemic, given environmental fluctuations and vaccine introduction.

Over 340 million people have been infected with SARS-CoV-2 since its discovery in late 2019[1]. In response, the scientific community has come together in an unprecedented effort to understand the epidemiology and biological mechanisms driving COVID-19. Effective vaccines are crowning achievements stemming from this effort. While these vaccines greatly reduce morbidity and mortality, vaccinated patients can still get infected and severe disease is possible[2]. The importance of therapeutics to counter these breakthrough infections is clear. Genetic studies have the potential to contribute to drug discovery by highlighting key biological pathways and potential therapeutic targets[3].

The most recent meta-analysis by the COVID-19 Host Genetics Initiative (HGI) identified 21 loci associated with susceptibility and 40 loci associated with COVID-19 hospitalization. These loci accounted for 1.2% and 5.8% of phenotypic variation respectively[4]. The case to invest additional time and resources into more genetic studies would be bolstered with evidence that a sizeable portion of the genetic architecture of susceptibility and severity is yet to be discovered. Heritability studies based on family pedigree information can provide such

evidence. These studies estimate narrow-sense heritability, which is the proportion of phenotypic variation attributed to additive genetic factors. These studies are not constrained by the kinds of loci discoverable using GWAS and instead take advantage of the natural genetic overlap between relatives (i.e. parents share 50% of genetic material with their children, 25% with their grandparents etc.)[5].

Pedigree based models have been used to estimate the narrow-sense heritability of other infectious disease phenotypes including susceptibility to developing malaria[6], tuberculosis[7,8], and Helicobacter pylori infection[9]. For COVID-19, a twin study by Williams et al. estimated that heritability of COVID-19 susceptibility was approximately 31%. This study relied on survey data and did not explore severity[10]. Heritability estimates where SARS-CoV-2 infection is determined by PCR testing would allow continuous, real-time heritability estimation.

In this study we estimate the shared environment and heritability of SARS-CoV-2 susceptibility and COVID-19 severity using electronic health record (EHR) data from New York-Presbyterian/Columbia University Irving Medical Center (NYP/CUIMC) and linked pedigree data[11].

[1]Department of Biomedical Informatics, Columbia University, New York, NY, USA. [2]Department of Physiology & Cellular Biophysics, Columbia University, New York, NY, USA. [3]Department of Computational Biomedicine, Cedars-Sinai Medical Center, West Hollywood, CA, USA. [4]Cedars-Sinai Cancer, Cedars-Sinai Medical Center, Los Angeles, CA, USA. ✉e-mail: nicholas.tatonetti@cshs.org

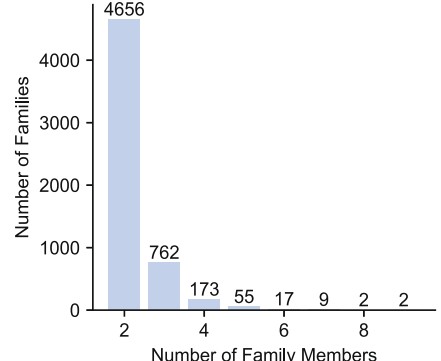

| Demographics | Cases | Controls |
|---|---|---|
| Patients | 1,580 | 11,184 |
| Female (%) | 1091 (69.05%) | 7463 (66.73%) |
| Age Cases (Std. Dev) | 46.3 (25.1) | 38.9 (25.3) |
| **Race/Ethnicity** | | |
| Black (%) | 192(1.5%) | 1226 (9.6%) |
| Hispanic (%) | 348 (2.7%) | 2180 (17.1%) |
| Other (%) | 882 (6.9%) | 5730 (44.9%) |
| White (%) | 158 (1.2%) | 2048 (16.1%) |

**Fig. 1 | Sample demographics and family size information. a** Summary demographics for SARS-CoV-2 PCR tested individuals with family information at NYP/CUIMC. **b** Number of members in each family with SARS-CoV-2 PCR results.

We identify case and control status using PCR test results and tracked patient hospital use. We test several approaches for defining shared environment in COVID-19 and estimate point heritability and shared environment estimates for SARS-CoV-2 susceptibility and COVID-19 severity phenotypes. We find that shared environment and genetic variation explain a moderate proportion of patient variation for both severity and susceptibility phenotypes. We are the first study to our knowledge to further examine how heritability and shared environment estimates change over the course of the pandemic by estimating them on a weekly basis, highlighting the challenges in modeling stable estimates in a changing environment. We find that over time heritability of SARS-CoV-2 susceptibility increased while the estimates for shared environment decreased.

## Results

We identified 12,764 patients in our pedigree that received a conclusive (positive or negative) PCR test for SARS-CoV-2 (Fig. 1). These patients belonged to 5,676 families with an average of 2.5 SARS-CoV-2 tested members per family (Fig. 1).

### Selecting shared environment

We considered three possibilities for modeling shared environment: grouping by family, household, or building. We assumed there is less variation in the temporality of positive SARS-CoV-2 tests in patients with similar exposure, or in other words, that the timing of infection for patients in a shared environment will be correlated. Therefore, the optimal shared environment modeling strategy would be the one that showed the least variation in the time it takes from one positive test within the group to the next. We compared the mean time to subsequent positive COVID tests for each group (Fig. 2) and used an ANOVA test to identify differences. We then performed a post-hoc analysis using individual two-sided t-tests. We found that the mean time to subsequent positive test was 33 days, 62 days, and 94 days for household, family, and building, respectively (Fig. 2). These differences were significant in an ANOVA ($F = 11.77$, $P = 9.42 \times 10^{-7}$) and household was significantly different from family ($T = 2.56$, $P = 0.01$, 95% CI = 5.49–51.99) and building ($T = 5.21$, $P = 3.22 \times 10^{-7}$, 95% CI = 34.95–85.08).

### Susceptibility estimates

We estimated the heritability of COVID-19 susceptibility to be 65% (95% CI: 33–80%) and shared environment to be 35% (95% CI: 15–51%) (Table 1). Cases were those patients that had at least one positive PCR test for SARS-CoV-2 and controls were patients with at least one negative PCR test and no positive PCR tests. These results were largely stable to excluding the proband requirement and across different sample sizes (Supplementary Table 1, Supplementary Fig. 1). Age has a greater effect on susceptibility compared to sex as seen in

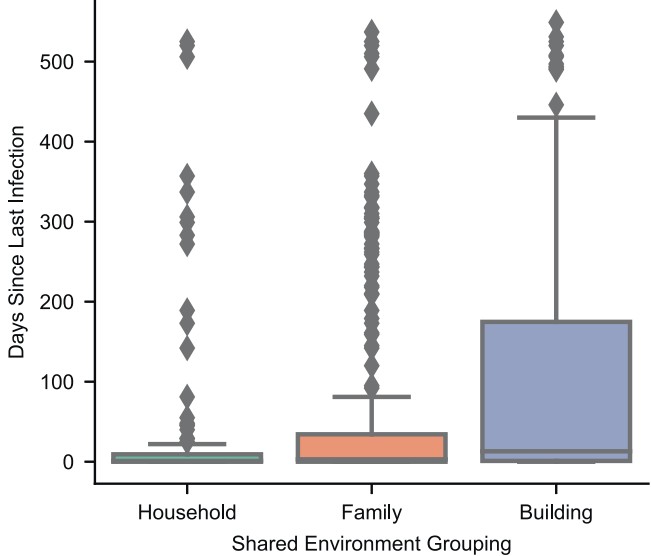

**Fig. 2 | Days to subsequent positive SARS-CoV-2 PCR tests by shared environment grouping.** Boxplot showing the days to subsequent positive SARS-CoV-2 PCR tests when grouping patients by household, family, and building. The center line is the median and the box limits are the first and third quartiles. The end points are 1.5x the inter-quartile range. Each individual group (i.e., Household, Family, or Building) must have a minimum of 2 PCR positive patients to be able to calculate time to next positive infection. $n = 269$ patients for Household, $n = 485$ patients for Family, and $n = 519$ patients for Building. Source data are provided as a Source Data file.

the change in heritability and shared environment estimates when we exclude these covariates individually and together (Supplementary Fig. 2). We control for age and sex differences in the main part of the manuscript given the consistent discrepancy in average age between our case and control groups, as well as the variable impact of age on vaccine response (Supplementary Fig. 3).

First degree relatives were more likely to live in the same household and most patients were first-degree relatives (Fig. 3a, b), potentially confounding our estimates. To evaluate our ability to differentiate shared environment from genetics, we estimated heritability and shared environment using only those families with higher degree relatives (second, third, and fourth degrees). For this sample, we found that genetics explained 57% (95% CI: 37–78%) of variation while shared environment explained 34% (95% CI: 20–49%) (Table 1). Narrow-sense heritability estimates and shared environmental estimates do not necessarily sum to 1, given the unique environment and error term included in the model.

**Table 1 | Heritability and shared environment estimates**

| Category | Trait | Narrow-sense heritability estimate | Shared environment estimate | Families | Quality score |
|---|---|---|---|---|---|
| Susceptibility | +PCR | 65% (95% CI: 33–80%) | 35% (95% CI:15–51%) | 1324 | 0.84 |
| | +PCR High Degree Families | 57% (95% CI: 37–78%) | 34% (95% CI:20–49%) | 296 | 0.86 |
| | Reinfection | Insufficient Sample | | 17 | – |
| Severity | Hospitalization | 55% (95% CI: 51–57%) | 45% (95% CI:43–49%) | 115 | 0.03 |
| | Days Hospitalized | 41% (95% CI: 33–57%) | 33% (95% CI:24–38%) | 158 | 0.9 |
| | +PCR w/ Intubation | Insufficient Sample | | 26 | |
| | +PCR w/ Death | Insufficient Sample | | 0 | |

Highest quality heritability and shared environment results as determined by the Quality Score and sample size. A high-quality estimate has a Quality Score greater than or equal to 0.8. Poor quality estimates are those with Quality Scores ≤0.2 (see Polubriaginof et al.'s discussion of POSA)[11]. No estimate means no phenotypic variance was accounted for. Binary traits were modeled with (default) and without a proband (case) requirement for family inclusion. (See Polubriaginof et al.)[11] Narrow-sense heritability estimates and environmental estimates do not necessarily sum to 1, given the unique environment and error term included in the model.

a)

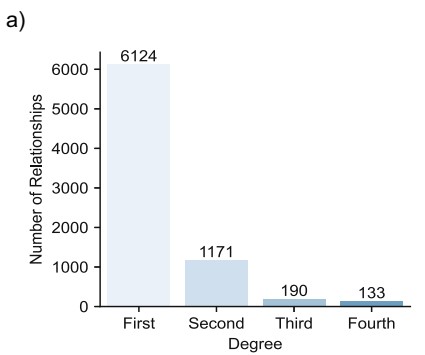

b)

| Degree | Sample Size | Shared HHID | Percent |
|---|---|---|---|
| First | 6124 | 3294 | 54% |
| Second | 1171 | 255 | 22% |
| Third | 190 | 13 | 7% |
| Fourth | 133 | 9 | 7% |

**Fig. 3 | Family composition and household information. a** Number of first-, second-, third-, and fourth-degree relationships. Patients with Unknown or Unspecified relationships excluded. **b** Percent relationships that share a household.

HHID refers to household ID, an identifier shared by patients that are family members living at the same address.

### Time varying heritability estimates

Changes in local prevalence, implementation of individual risk mitigation factors, and access to vaccination likely affect an individual's risk tied to their shared environment in a meaningful way. To examine this more closely, we estimated the heritability and shared environment at weekly, cumulative intervals starting one week after the first case. For each date cutoff, cases were those patients that received a positive PCR test before or on that date. Controls were all patients who tested negative prior to the date or that have any PCR test, positive or negative, in the future. We included patients that in the future have a positive test as controls because we operated under the potentially flawed assumption that such patients were negative for SARS-CoV-2 at that prior time point. The relative importance of shared environment compared to genetics shifted over the course of our study period. Heritability estimates ranged from 33% (95% CI: 20–46%) during the first half to 70% (95% CI: 63–74%) in the second half (Fig. 4, Supplementary Fig. 4).

### Heritability estimates COVID-19 severity

We used hospitalization with COVID-19 and days hospitalized while infected with COVID-19 as proxies for COVID-19 severity. Estimates for hospitalization status were not stable, with Quality Scores well under the pre-defined thresholds (*Methods*) (Table 1). For days hospitalized, we found that shared environment accounted for 33% (95% CI: 24–38%) of variation while additive genetic variation accounted for 41% (95% CI: 31–52%) (Table 1). Days hospitalized estimates were stable at different sample sizes (Supplementary Fig. 1). Controlling for sex and age had little effect on our models, as seen in the stability of the estimates when excluding these covariates (Supplementary Fig. 2).

### Permutation analysis

To gain confidence that patient genetic variation and shared environment explain phenotypic differences between individuals we

permuted patient SARS-CoV-2 status and days hospitalized and re-estimated the contributions of patient genetic variation and shared environment to these phenotypes (Supplementary Methods). As expected, no phenotypic variation was attributed to additive genetic variation and shared environment for either permuted phenotype at any tested sample size (20–90%).

## Discussion

We estimated the heritability and shared environment of SARS-CoV-2 susceptibility and COVID-19 severity traits. Our high heritability estimates suggest the genetic architecture of susceptibility and severity have not been fully uncovered, though the shifting dynamics of the pandemic make identifying steady-state heritability and shared environment estimates challenging. Changes in prevalence, access to testing and high-quality masks, quarantining, and social distancing practices all impacted a patient's environment[12] in ways that likely affected the estimated importance of shared and personal environment. Additionally, as testing options increased, our confidence in patient control status decreased since patients likely took subsequent positive tests outside of the NYP/CUIMC system[13]. This is further complicated by the fact that the virus itself mutated and vaccines became available, potentially altering the importance of different genetic and environmental factors.

We see the impact of these shifts reflected in our cumulative analysis where we estimated the importance of shared environment and heritability to susceptibility on a weekly basis. We found that as the pandemic progressed, the relative importance of shared environment and additive genetic variation flipped, with additive genetics accounting for a greater proportion of phenotypic variation in the second half of our study. While this study cannot pinpoint the drivers of these changes, we expect that the homogenization of shared environment played some role. Over time, the probability that no

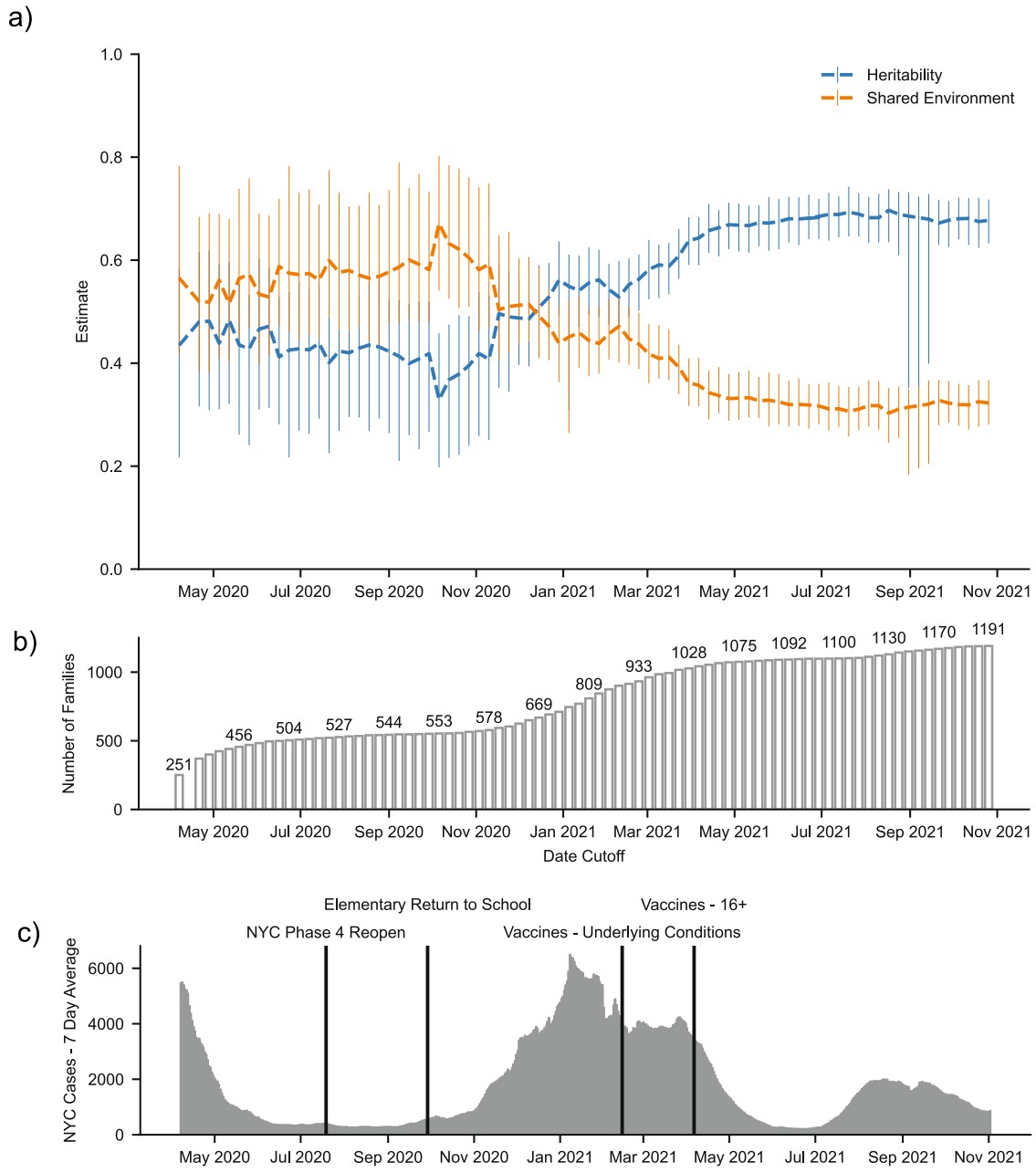

**Fig. 4 | Heritability and shared environment estimates over time for SARS-CoV-2 susceptibility. a** Cumulative heritability and shared environment estimates with 95% CI. Center points are the heritability and shared environment estimates from the model associated with the median, significant heritability estimate. *n* = 3174 patients. Source data are provided as a Source Data file. **b** Bar chart shows the number of families in each analysis. **c** Bar chart with daily 7 day average cases in New York City[23]. Highlighted date points include New York City's Phase 4 re-opening, the return of elementary school students to school, the rollout of vaccines to those with underlying conditions, and the rollout of vaccines to the broader population 16 years and older.

household member has been infected decreased as public health measures promoting isolation, such as New York's On Pause program, rolled back and patients returned to normal routines. Additionally overall access to risk mitigating factors such as high-quality masks and testing increased. These changes likely reduced inter-patient differences in shared environment. This homogenization of risk would naturally inflate the importance of genetic differences between the families. The genetic pools of our case and control groups also likely changed in meaningful ways. Patients with protective alleles may have accounted for a higher proportion of our control group over time as

patients with risk and neutral alleles were infected and moved to the case group. Lastly, we expect any genetic factors affecting COVID-19 vaccine efficacy to impact our estimates with their introduction partway through our study.

To our knowledge no other study has examined the change in heritability and shared environment estimates for susceptibility over time. A related study by Warmerdam et al. found the predictiveness of polygenic risk scores for survey-reported SARS-CoV-2 infection decreased as the pandemic progressed. It is hard to determine if the importance of genetics overall changed for this population or if the

variants and specific effect sizes used to calculate the polygenic risk scores became less informative as time went on. Additionally, the variants and effect sizes used provide limited information, accounting for less than 1% of phenotypic variation in the original COVID-19 Host Genetics Initiative (HGI) study[14,15]. The dynamic nature of our susceptibility estimates cautions against putting too much confidence into any single heritability estimate derived during the pandemic and highlights the difficulties in between study comparisons of such estimates. Williams et al.'s survey-based twin study estimated the heritability of susceptibility at 31%[10]. Restricting our dataset to the same time frame we similarly estimate heritability at 37% with 63% for shared environment. Compared to our end of study estimate though, we found a very different allocation of variance with heritability at 65% and shared environment at 35%.

For every time cut, our susceptibility heritability estimate is substantially greater than the HGI estimate of 1.2%. Similarly, our days hospitalized estimate was higher than HGI's 5.8% estimate for hospitalization due to COVID-19 and 8.2% estimate for critical illness[4].

While differences in our sample populations likely account for some of these differences, variation in phenotype definition and modeling approach also likely play a role. Only some of the studies included in the HGI meta-analysis studies used PCR tests to define susceptibility as we did. Additionally, HGI used hospitalization due to COVID-19 and hospitalization with poor outcomes, such intubation and death, as severity traits[4,15]. In contrast we were limited to using days hospitalized and hospitalization status based on patients hospitalized with COVID-19 instead of due to COVID-19. Cause of hospitalization is difficult to discern systematically. We try to limit this effect by requiring the patient's positive SARS-CoV-2 test to occur 12 days before hospitalization or within the first 96 h of hospitalization. Despite this, our sample likely includes patients hospitalized for other causes given the standard to test all patients upon admittance. Only 17 families in our pedigree had patients that were intubated, which prohibited a severe hospitalization analysis.

There are also trade-offs between HGI's SNP based modeling approach and our pedigree-based approach. Imperfect tagging of causal variants, particularly rare and structural variants, can cause SNP based methods to underestimate trait heritabilties[5,16,17]. Additionally, common variants with small effect sizes may not be adequately estimated. While shared environment is generally not a confounder in SNP-based heritability studies, heritability may be biased if population stratification is correlated with genetic effect[17]. Given the regional differences in COVID prevalence and risk mitigation strategies, environment may still bias these results.

In contrast, pedigree based methods, like what we use here, are prone to over-estimating heritability due to epistasis and mis-attribution of what should be shared or personal environment[16]. We expect shared environment is particularly challenging to differentiate in our study, given its intrinsic importance in susceptibility to infectious disease, the complexity of defining it, the overlap between shared environment groupings and family members, and the limited number of complex families in our sample (families with second or third degree relatives).

Environment influences susceptibility and severity in two apparent ways: indirectly by adjusting the risk of relevant comorbidities[18] and directly by influencing a patient's viral exposure. Relevant comorbidities can be affected by a patient's environment, which overlaps with family members to varying extents. Likely more important for susceptibility is overlap in shared viral exposure between patients. We considered three potential groupings for defining shared environment to capture shared viral exposure: a patient's immediate household, their family, and their building. Since many patients in our sample live in apartment buildings or assisted living facilities, household and building groupings were often not equivalent. We assumed there is less variation in the temporality of positive SARS-CoV-2 tests in patients with similar exposure. We found that the household level had the lowest variation in time between tests and used this to define shared environment. The intrinsic overlap between family members and household members makes differentiating between heritability and shared environment challenging[19]. To gain confidence in our estimate, we re-ran our analysis excluding families with only first-degree relatives and found similar results.

Still, non-genetic patient factors that follow family lines may have inflated our heritability estimates. A challenge study by Killingley et al. found that only 53% of their subjects directly exposed to SARS-CoV-2 developed COVID-19. While the authors were unable to determine why some patients did not develop COVID-19, they speculated that they may have cross re-reactive immune cells that provided protection[20]. We would expect that patient-specific, risk mitigating factors should become easier to identify as the pandemic progresses and patients that lack these factors are infected. Whether these differences are due to genetic variation, use or response to vaccines, or variation in immune cell population is difficult to untangle. If these differences are more likely to follow family-lines, we would expect the variation to be attributed to heritability in our study, which could inflate our heritability estimates.

Our study highlights the challenges in estimating heritability and shared environment over the course of a pandemic. Observational datasets, such as ours, allow for quick, continual re-estimation of heritability as relevant dynamics change. As discussed, our heritability estimates are likely inflated due to modeling methodology, family structure sample constraints, and confounding variables that may follow family lines. Still, taken together with the survey-based estimates by Williams et al[10]. and Averitt et al[21]., our results suggest that the genetic architectures for SARS-CoV-2 susceptibility and COVID-19 severity have not been fully uncovered. Additional genetic studies using alternative study designs and whole genome sequencing should be pursued to further uncover the drivers of inter-patient variation in SARS-CoV-2 susceptibility and COVID-19 severity. Results from such studies have the potential to inform future drug development and patient risk-stratification[3,22]. Our over-time estimates of susceptibility heritability and shared environment reflect the shifting dynamics of the pandemic and cautions against the generalization of any one study. Future studies estimating heritability and shared environment across multiple sites and time points should be conducted to determine steady-state heritability and shared environment population estimates as COVID-19 enters a fully endemic state, with stabilized public health measures. These estimates are never expected to be fully uniform or stable though, given their population specific nature. Public health agencies can take heritability and shared environment estimates into account when crafting new policies.

## Methods

This study is approved by the Columbia University Institutional Review Board (#AAAL0601).

### Data collection and cohort selection

Patients from the NYP/CUIMC data warehouse were included if they had a recorded SARS-CoV-2 PCR test between February 21, 2020, and October 24, 2021, and were in our pedigree. Families were only included if there were at least two patients with PCR results as required by SOLARStrap. NYP/CUIMC data warehouse data is not publicly available due to legal regulations.

### Phenotype definitions: SARS-CoV-2 susceptibility

Patients were considered a case if they had a positive SARS-CoV-2 PCR test and were a control if they had at least one negative SARS-CoV-2 PCR test and no positive results. We excluded controls with ICD codes indicating previous SARS-CoV-2 infection.

## Phenotype definitions: COVID-19 severity

For hospitalization status patients were considered a case if they were admitted within 12 days of a positive test or tested positive within the first 96 h of a hospitalization. A patient was considered a control if they tested positive outside of a hospitalization and not within 12 days of the start of a hospitalization. For patients that were hospitalized, days hospitalized were calculated from the start of the encounter to discharge. If the patient was re-admitted within 5 days of discharge, the number of days was extended to the end of the subsequent encounter. Patients that were not hospitalized were assigned zero days.

## Shared environment definition and selection

Building ID was determined using patient address data. The ID comprised of the patient's zip code, street, and street number. Family grouping was determined using the Family ID from the pedigree data. Address data was too irregular to pull apartment level data for patients. To circumvent this, we assumed that patients within the same family that live in the same building likely live in the same apartment. Household ID was assigned by concatenating the Building ID with the Family ID.

We assumed that patients that inhabit the same environment would become infected around the same time since the initial infectious patient would expose surrounding members soon after their own infection. Using this intuition, we compared the average time to subsequent positive infection when we grouped patients by building, family, and household using an ANOVA model followed by individual two-sided student's $t$-tests to determine the best representation for shared environment.

## Heritability and shared environment estimation

We used SOLARStrap to estimate the heritability and shared environment for SARS-CoV-2 susceptibility and severity metrics. SOLARStrap was created specifically to help control for ascertainment bias in observational datasets through repeated subsampling. It was previously validated against 91 literature derived heritability estimates, and was demonstrated to be robust to both biased (nonrandom missingness) data and missing at random data through simulations[11]. Prior implementations of SOLARStrap did not provide a total shared environment estimate, though it included shared environment in individual ACE models. These underlying models are variance-component models which partition the outcome variance into three groups: additive genetic effects (A), common environment (C), and unique environment and error (E). In this implementation, we report the overall shared environment estimate as the shared environment from the model with the median significant heritability estimate (see SOLARStrap release version 1.0.0). We set SOLARStrap to run 200 iterations of SOLAR using between 20% and 90% of available families, up to 3000 total families, SOLARStrap's maximum (see model sensitivity analysis below) and included sex and age as covariates. SOLARStrap automatically normalizes quantitative variables using an inverse normal transformation.

To indirectly explore the effects of age and sex on our outcomes, we ran the analyses three additional times controlling for only age, only sex, and neither age nor sex. The magnitude of the changes in heritability and shared environment estimates point to the importance of the various covariates.

To understand how the heritability and shared environment estimates changed over the course of the pandemic we estimated the heritability at weekly cumulative date cuts, including 90% of families because of small sample size early in the pandemic. We required a minimum of 40 families to run the analysis. Patients were a case if they had a positive PCR test by the date cutoff and were a control if they had any PCR test at any point during the study and were not a case at that point. This allowed class switching between weeks.

## Model sensitivity

To understand the sensitivity of our results to sample size we estimated shared environment and heritability 8 times for each phenotype, increasing our sample size by increments of 10% of available families starting at 20%. For example for the first estimation, SOLARStrap ran 200 iterations of SOLAR with a randomly selected subset of 20% of available families (see Polubriaginof et al. for additional background on SOLARStrap)[11]. We repeated this process excluding a proband (case) requirement for family inclusion for dichotomous phenotypes to check estimate stability for this modeling choice. As recommended by previous work, we selected estimates derived from the fewest number of families with a Quality Score of ≥0.8 (see Polubriaginof et al.'s discussion of POSA)[11].

## Complex family analysis

To gain confidence in our ability to differentiate shared environment from heritability we estimated the heritability and shared environment of families that contained higher degree relatives (second-, third- and fourth-degree relatives) since they were less likely to live together.

## Software

We conducted our analyses using Python version 3.8.10, MySQL version 5.6, and SOLARStrap version 1.0.0. SOLARStrap is available at https://github.com/tatonetti-lab/h2o.

## Reporting summary

Further information on research design is available in the Nature Portfolio Reporting Summary linked to this article.

# Data availability

We used electronic health records from NYP/CUIMC's data warehouse which are protected in the United States from public access through the 1996 Public Law 104-191 (HIPAA). NYP/CUIMC policy does not allow for the release of patient-level data. The underlying data for the figures can be found in the Source Data file. Source data are provided with this paper.

# Code availability

Python scripts used for data selection, cleaning, and graph creation are available on GitHub at https://github.com/tatonetti-lab/covid-h2o. DOI: 10.5281/zenodo.10223312.

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

## Acknowledgements
K.L.B., M.Z., V.R., U.G., and N.P.T are funded by NIH NIGMS R35GM131905.

## Author contributions
K.L.B. and N.P.T. designed the study. V.R. worked on data acquisition. K.L.B., M.Z., and U.G. worked on data processing. K.L.B. carried out the analyses with input from N.P.T. K.L.B. and N.P.T. wrote, revised, and approved the final version of the manuscript.

## Competing interests
The authors declare no competing interests.
