## [Peer Review File · Nature Communications]

Estimating the Heritability of SARS-CoV-2 Susceptibility and COVID-19 SeverityREVIEWER COMMENTS

Reviewer #1 (Remarks to the Author):

The authors study the heritability of SARS-CoV-2 susceptibility using pedigree information in the New York area. The authors conclude that at the beginning of the pandemic the estimated heritability is 33% in the 2020, and later going up to 70% in 2021. This is an intriguing observation, yet to me these results are also somewhat confusing: I would be inclined to think that the estimated heritability of SARS-CoV-2 susceptibility should have diminished over time, since the majority of people at some point got infected (although I realise the authors have studied here only the years 2020 and 2021): the amount of times that an individual was exposed goes up with time, and even if you are genetically somewhat protected, this will still cause you to get infected at some point, provided you get exposed sufficiently often.

This is also observed in a paper that studied how the influence of genetic variation has changed over time (doi:10.1371/journal.pgen.1010135), and observed that the explanatory power of a polygenic risk score for SARS-CoV-2 susceptibility, constructed using GWAS summary statistics from the HGI, decreased over the course of the pandemic, which I believe is in contrast to what the authors describe here.

I am also somewhat confused when the authors compare their findings to the results of the COVID19 Host Genetics Initiative (HGI), where the estimated heritability was <1%. I wonder, could it be that the heritability estimates could be somewhat too high?

Regretfully my knowledge on the software that the authors used here (SOLARStrap) is very limited, but I worry somewhat that the software might have issues with dealing with censored or time-to-event data (which I would consider the input data is here). The authors spend few words on the method, and do not mention how they have accounted for this, and how the authors have ensured that the software can handle this type of data. For instance, as a sanity check, can the authors use the same software and dataset to estimate the heritability of height and of for instance age? Does the software then conclude that these are ~80% and 0%, respectively? And can the authors simulate some censored type of data with known heritability? I would be very important to carefully consider this, and also to study what the effect of loss-to-follow-up is.

I would appreciate it a lot if the authors would be willing to spend efforts to check the assumptions behind the SOLARStrap model and run simulations with time-to-event data.

Minor comments:

"estimated that heritability of COVID-19" should read "estimated that
56 heritability of COVID-19 susceptibility"

"likely effect an individual" should read "likely affect an individual"

Figure 1a: "Cummulative" should read "Cumulative"

Reviewer #2 (Remarks to the Author):

The manuscript titled "Estimating the Heritability of SARS-CoV-2 Susceptibility and COVID-19 Severity" deals with the individual and environmental factors that contribute to the susceptibility and response to SARS-CoV-2 infection.

The authors aimed to estimate the heritability to SARS-CoV-2 susceptibility and COVID-19 severity analysing a sample of about 12,500 patients and using hospitalization as a surrogate of clinical aggravation. Statically significant estimates were found for most of the variables examined and a high to moderate heritability was detected (Covid-19 susceptibility from 57% to 67% and severity of the disease from 41% to 55%). In addition, the authors investigated the heritability to SARS-CoV-2 susceptibility and the role of shared environmental factors over the course of the pandemic.

The manuscript seems to me rather interesting and, in my opinion, it is well written and well organized. Therefore, I recommend the acceptance of this manuscript after accommodating the following minor comments.

- The Covid-19 pandemic is affecting almost the entire world population; however, epidemiological data showed that age and sex play a critical role. Could the authors provide information on age and sex effect? This information is important to better understand the impact of the covariates on genetic and environmental components.

- Authors declared that age and sex were included as covariates. I would suggest that the authors perform a stratified analysis by sex rather than an analysis adjusted by sex. Understanding the role of sex in moderating susceptibility to Covid-19 could further enrich this research and support the authors' theory.

- The author stated that they identified 12,757 patients (Line 70) but in Figure 1 the number of patients reported is 12,549. Could the authors explain why?

- Line 71 - Results section, "Table 1", should be Figure 1

- Figure 1a shows demographic characteristic of all the patients. To make this information more complete and exhaustive, the authors should report these data broken down by cases and controls (positive and negative to PCR test, respectively).

- The author estimated the proportion of phenotypic variation attributed to Narrow-sense heritability and Shared Environment (i.e. + PCR heritability=65% and Shared environment=35%, based on a total of 100%). The authors should explain better why in the case of "+ PCR High Degree Families" and "Days Hospitalized" the total does not add up to 100%.

- Line 134/135, - Results section, 41% of the variation refers to the Narrow-sense Heritability Estimate. Authors should fix this error.

- Figure 3b - Results section, the authors need to add in the note an explanation of "Shared HHID".

- Line 282 - Discussion Session, "ACE", this acronym should be explained.

- The article would benefit from a greater discussion of the fallout of this research in public health practice and clinical medicine. It would be important to clarify the implications of the results and the likely direction of future work.

Reviewer #1 (Remarks to the Author):

The authors study the heritability of SARS-CoV-2 susceptibility using pedigree information in the New York area. The authors conclude that at the beginning of the pandemic the estimated heritability is 33% in the 2020, and later going up to 70% in 2021. This is an intriguing observation, yet to me these results are also somewhat confusing: I would be inclined to think that the estimated heritability of SARS-CoV-2 susceptibility should have diminished over time, since the majority of people at some point got infected (although I realise the authors have studied here only the years 2020 and 2021): the amount of times that an individual was exposed goes up with time, and even if you are genetically somewhat protected, this will still cause you to get infected at some point, provided you get exposed sufficiently often.

Thank you, we understand your prior is that heritability estimates should decrease over the course of the pandemic since patients with protective variants will eventually become infected and move from control to case group. We think what is happening is three-fold, though demonstrating these is not within the scope of our analysis:

- 1) Homogenization of environment - early on there was more variable opportunity for viral exposure and risk-mitigating factors (access to masks, access to testing, access to space to quarantine, essential vs. non-essential workers etc.). We believe that over time, inter-family differences in these areas decreased as public health measures lifted, access to risk-mitigating factors increased, and people returned to “normality”. The resulting decrease in inter-patient shared environment differences would intrinsically increase the proportion of variation additive-genetics accounts for.

- 2) Case-control group genetic makeup –Over time the case group may purify as patients with neutral/risk variants are infected. As you point out though if protective variants aren't purely protective, patients will eventually move to the case group and we would have a harder time picking out the genetic signal. We may see this if we were to continue the study, although our confidence in control status based on hospital-administered PCR tests will decrease as at-home rapid tests became available.

- 3) Vaccines - Any genetically driven differences in vaccine efficacy will impact our estimates part-way through our study.

We slightly modified our discussion to clarify these points in the manuscript.

This is also observed in a paper that studied how the influence of genetic variation has changed over time (doi:10.1371/journal.pgen.1010135), and observed that the explanatory power of a polygenic risk score for SARS-CoV-2 susceptibility, constructed using GWAS summary statistics from the HGI, decreased over the course of the pandemic, which I believe is in contrast to what the authors describe here.

Thank you for pointing us to this study. We think it's hard to tell if the specific variants and effect sizes used to calculate the PGS became less predictive over time for this population, or if genetics overall became less important. Additionally, we expect the variants and effect sizes used to calculate the polygenic risk scores provide limited information for this population given that they account for <1% in the original HGI study. We've added a brief discussion about this study in our manuscript.

I am also somewhat confused when the authors compare their findings to the results of the COVID19 Host Genetics Initiative (HGI), where the estimated heritability was <1%. I wonder, could it be that the heritability estimates could be somewhat too high?

Yes, we think that both our method over-estimates the heritability and HGI's underestimates it and discuss trade-offs in these two approaches in our discussion. We separated out the paragraph discussing the likely over-estimation of our approach in the discussion to make these caveats more apparent and easier to find for the reader. We include the two paragraphs below for easier review:

"There are also trade-offs between HGI's SNP based modeling approach and our pedigree-based approach. Imperfect tagging of causal variants, particularly rare and structural variants, can cause SNP based methods to underestimate trait heritabilities.^{6,18,19} Using whole genome sequencing, the GenOMICC study was able to recover more of the predicted heritability for COVID-19 severity.⁵ Additionally, common variants with small effect sizes may not be adequately estimated. While shared environment is generally not a confounder in SNP-based heritability studies, heritability may be biased if population stratification is correlated with genetic effect.¹⁹ Given the regional differences in COVID prevalence and risk mitigation strategies, environment may still bias these results.

In contrast, pedigree based methods, like what we use here, are prone to over-estimating heritability due to epistasis and mis-attribution of what should be shared or personal environment.¹⁸ We expect shared environment is particularly challenging to differentiate in our study, given its intrinsic importance in susceptibility to infectious disease, the complexity of defining it, the overlap between shared environment groupings and family members, and the limited number of complex families in our sample (families with second or third degree relatives)."

We also touch on this when discussing potential confounders considering the challenge study by Killingley et al. (<https://doi.org/10.1038/s41591-022-01780-9>) Relevant portion below for ease of reference:

"Still, non-genetic patient factors that follow family lines may have inflated our heritability estimate. A recent challenge study by Killingley et al. found that only 53% of their subjects directly exposed to SARS-CoV-2 developed COVID-19. While the authors were unable to determine why some patients did not develop COVID-19, they speculated that they may have

cross re-reactive immune cells that provided protection.²² We would expect that patient-specific, risk mitigating factors should become easier to identify as the pandemic progresses and patients that lack these factors are infected. Whether these differences are due to genetic variation, use or response to vaccines, or variation in immune cell population is difficult to untangle. If these differences are more likely to follow family-lines, we would expect the variation to be attributed to heritability in our study, which could inflate our heritability estimates.”

Regretfully my knowledge on the software that the authors used here (SOLARStrap) is very limited, but I worry somewhat that the software might have issues with dealing with censored or time-to-event data (which I would consider the input data is here)

The authors spend few words on the method, and do not mention how they have accounted for this, and how the authors have ensured that the software can handle this type of data. For instance, as a sanity check, can the authors use the same software and dataset to estimate the heritability of height and of for instance age? Does the software then conclude that these are ~80% and 0%, respectively?

And can the authors simulate some censored type of data with known heritability? I would be very important to carefully consider this, and also to study what the effect of loss-to-follow-up is. I would appreciate it a lot if the authors would be willing to spend efforts to check the assumptions behind the SOLARStrap model and run simulations with time-to-event data.

Thank you for raising these concerns. We have updated our manuscript to highlight that this is a previously published and validated method. SOLARStrap was validated by Polubriaginof et al. (doi: 10.1016/j.cell.2018.04.032) using EHR from 3 separate sites. In this study the authors compared 91 SOLARStrap derived heritability estimates to literature ones. They found that SOLARStrap estimates were significantly correlated with literature ones. For height specifically, the SOLARStrap heritability estimate was 0.80 (95% CI: 0.74–0.86).

Loss-to-follow-up is fundamentally a problem of missing data. Polubriaginof et al. demonstrated SOLARStrap’s robustness to missing data by re-capitulating heritability estimates for a simulated phenotype when varying degrees of missingness were introduced to the pedigree and phenotypic data.

Minor comments:

"estimated that heritability of COVID-19" should read "estimated that 56 heritability of COVID-19 susceptibility"

Thank you, we have addressed this.

"likely effect an individual" should read "likely affect an individual"

Thank you, we have addressed this.

Figure 1a: "Cummulative" should read "Cumulative"

Thank you, we have addressed this.

Reviewer #2 (Remarks to the Author):

The manuscript titled “Estimating the Heritability of SARS-CoV-2 Susceptibility and COVID-19 Severity” deals with the individual and environmental factors that contribute to the susceptibility and response to SARS-CoV-2 infection.

The authors aimed to estimate the heritability to SARS-CoV-2 susceptibility and COVID-19 severity analysing a sample of about 12,500 patients and using hospitalization as a surrogate of clinical aggravation. Statically significant estimates were found for most of the variables examined and a high to moderate heritability was detected (Covid-19 susceptibility from 57% to 67% and severity of the disease from 41% to 55%). In addition, the authors investigated the heritability to SARS-CoV-2 susceptibility and the role of shared environmental factors over the course of the pandemic. The manuscript seems to me rather interesting and, in my opinion, it is well written and well organized . Therefore, I recommend the acceptance of this manuscript after accommodating the following minor comments.

- The Covid-19 pandemic is affecting almost the entire world population; however, epidemiological data showed that age and sex play a critical role. Could the authors provide information on age and sex effect? This information is important to better understand the impact of the covariates on genetic and environmental components.

Thank you, we re-estimated heritability and shared environment for the phenotypes excluding age, sex, and both age and sex. Excluding age greatly affected the susceptibility estimate while sex did not. Given the discrepancy in ages between the case and control group and the effect of age on vaccine efficacy, we continued to control for age in the main reported estimates. We did not see a large effect by excluding sex and age for COVID-19 severity. We included these results in the supplementary files.

- Authors declared that age and sex were included as covariates. I would suggest that the authors perform a stratified analysis by sex rather than an analysis adjusted by sex. Understanding the role of sex in moderating susceptibility to Covid-19 could further enrich this research and support the authors’ theory.

As a pedigree-based method, we think running SOLARStrap in a sex-stratified manner is not recommended since pedigrees intrinsically contain males and females and SOLARStrap links individuals of varying genetic relatedness. Still, we tried estimating heritability and shared environment for susceptibility and severity in a sex-stratified manner out of curiosity. We were not able to get quality scores for susceptibility. For COVID-19 severity we see similar estimates

for females only and non-females only. We've included the graphs below. Given our reticence towards this approach for SOLARStrap, we decided not to include them in our manuscript.

- The author stated that they identified 12,757 patients (Line 70) but in Figure 1 the number of patients reported is 12,549. Could the authors explain why?

To look into, appreciate they caught this

Thank you for catching this. We incorrectly filtered on patients with certain relationships before calculating some of the demographic information. We have fixed this.

- Line 71 - Results section, "Table 1", should be Figure 1-

Thank you, we have addressed this.

- Figure 1a shows demographic characteristic of all the patients. To make this information more complete and exhaustive, the authors should report these data broken down by cases and controls (positive and negative to PCR test, respectively).

Thank you, we have broken out the demographic details by case and control status.

- The author estimated the proportion of phenotypic variation attributed to Narrow-sense heritability and Shared Environment (i.e. + PCR heritability=65% and Shared environment=35%, based on a total of 100%). The authors should explain better why in the case of “+ PCR High Degree Families” and “Days Hospitalized” the total does not add up to 100%.

Thank you, we have provided an explanation both in the text of the manuscript and under Table 1. Since this model includes three components: shared environment, heritability, and the error and personal environment term, shared environment and heritability will not always sum to 1.

- Line 134/135, - Results section, 41% of the variation refers to the Narrow-sense Heritability Estimate. Authors should fix this error.

Thank you, we have addressed this.

- Figure 3b - Results section, the authors need to add in the note an explanation of “Shared HHID”.

Thank you for pointing out this missing explanation. We have added an explanation of Shared HHID.

- Line 282 - Discussion Session, “ACE”, this acronym should be explained.

Thank you, we have added a brief description of ACE models.

- The article would benefit from a greater discussion of the fallout of this research in public health practice and clinical medicine. It would be important to clarify the implications of the results and the likely direction of future work.

Thank you, we expanded these points in the discussion.

REVIEWER COMMENTS

Reviewer #1 (Remarks to the Author):

I thank the authors for responding to the comments that I raised. I believe the authors provide reasonable explanations for the observed discrepancies with other research papers.

There is however one point that I do not agree with, and that is the loss-to-follow-up. The authors state in their response: "Loss-to-follow-up is fundamentally a problem of missing data. Polubriaginof et al. demonstrated SOLARStrap's robustness to missing data by re-capitulating heritability estimates for a simulated phenotype when varying degrees of missingness were introduced to the pedigree and phenotypic data."

While I agree that loss-to-follow-up is a problem data, it is a very particular problem, where missingness is non-random. The authors state that Polubriaginof et al. has demonstrated SOLARStrap's robustness to missing data, but this is not the same as my point.

Given the discrepancies with other literature, I do believe, it is valuable to consider as a potential explanation for the observations of this paper. As such I would like the authors to perform experiments on this, as I requested in my original review report.

Reviewer #2 (Remarks to the Author):

It seems that the authors have addressed most of my concerns.

One minor comments:

Line 145, 146, 170 and 171.

Heritability/environment estimates and related confidence intervals should both be expressed as a percentage (for example line 145: 57% (95% CI: 37%-78%) instead of 57% (95% CI: 0.37-0.78). Please check throughout the manuscript.

REVIEWER COMMENTS

Reviewer #1 (Remarks to the Author):

I thank the authors for responding to the comments that I raised. I believe the authors provide reasonable explanations for the observed discrepancies with other research papers.

There is however one point that I do not agree with, and that is the loss-to-follow-up. The authors state in their response: "Loss-to-follow-up is fundamentally a problem of missing data. Polubriaginof et al. demonstrated SOLARStrap's robustness to missing data by re-capitulating heritability estimates for a simulated phenotype when varying degrees of missingness were introduced to the pedigree and phenotypic data."

While I agree that loss-to-follow-up is a problem data, it is a very particular problem, where missingness is non-random. The authors state that Polubriaginof et al. has demonstrated SOLARStrap's robustness to missing data, but this is not the same as my point.

Given the discrepancies with other literature, I do believe, it is valuable to consider as a potential explanation for the observations of this paper. As such I would like the authors to perform experiments on this, as I requested in my original review report.

Thank you for raising your concerns. In the original paper we explored SOLARStrap's robustness to non-random missingness by removing pedigrees based on probabilities sampled from a beta distribution. The logic was that patients will have differing, non-random likelihoods of returning to a specific hospital and having a diagnosis for a phenotype. We found that traditional method (SOLAR) was sensitive to this non-random missingness and produced biased estimates (see figure below). In contrast, we found that SOLARStrap was robust to this non-random missingness. We found that excluding families in a non-random way still produced stable results using varying parameters for the beta distribution. The full description of this analysis and the relevant graphs are below for convenience. We have also updated the manuscript to mention this point, although we left out the details for brevity (also see below for ease of reference). We are happy to explore this issue further if you believe these simulations are insufficient.

[https://www.cell.com/cell/pdf/S0092-8674\(18\)30525-7.pdf](https://www.cell.com/cell/pdf/S0092-8674(18)30525-7.pdf)

Relevant section from the original paper:

"Evaluation of the robustness of SOLAR and SOLARStrap to biased data (non-random missingness)" To evaluate the robustness of SOLAR and SOLARStrap to biases, specifically non-random missingness, pedigrees were removed from the heritability estimation with a probability determined by a beta distribution. The beta distribution is a continuous probability distribution bounded by 0 and 1 and parameterized alpha and beta. Each family can be assigned a probability by sampling this distribution. Most families will have the same probability of missing data with a small number of families have a much lower probability. By varying the beta and alpha parameters we can change the proportion of families with a much lower probability

of missing data. We varied the value of the beta parameter from 0.001, 0.01, 0.1, 1.0, 10.0, to 100.0 and we set the alpha parameter such that the average probability of missingness across all families was constant at 50%.”

Figure Caption: SOLAR is sensitive to this bias and produces inaccurate results as the strength of the bias increases. SOLARStrap is robust to these biases and produces accurate estimates of heritability, even in the most extreme case of bias.

Updated section in the current manuscript:

“We used SOLARStrap to estimate the heritability and shared environment for SARS-CoV-2 susceptibility and severity metrics. SOLARStrap was created specifically to help control for ascertainment bias in observational datasets through repeated subsampling. It was previously validated against 91 literature derived heritability estimates, and was demonstrated to be robust to both biased (nonrandom missingness) data and missing at random data through simulations.¹²”

Reviewer #2 (Remarks to the Author):

It seems that the authors have addressed most of my concerns.

One minor comments:

Line 145, 146, 170 and 171.

Heritability/environment estimates and related confidence intervals should both be expressed as a percentage (for example line 145: 57% (95% CI: 37%-78%) instead of 57% (95% CI: 0.37-0.78). Please check throughout the manuscript.

Thank you for identifying these inconsistencies. We have fixed them.

REVIEWERS' COMMENTS

Reviewer #1 (Remarks to the Author):

The authors have convinced me about earlier reservations I had, by pointing out that the method is robust to loss-to-follow up.

As such I believe the paper is acceptable for publication.